# Post-Transcriptional Modifications of RNA as Regulators of Apoptosis in Glioblastoma

**DOI:** 10.3390/ijms23169272

**Published:** 2022-08-17

**Authors:** Anton Dome, Maya Dymova, Vladimir Richter, Grigory Stepanov

**Affiliations:** Institute of Chemical Biology and Fundamental Medicine of the Siberian Branch of the Russian Academy of Sciences, 630090 Novosibirsk, Russia

**Keywords:** glioblastoma, RNA modification, apoptosis, small nucleolar RNA, lncRNA, epigenetic regulation

## Abstract

This review is devoted to changes in the post-transcriptional maturation of RNA in human glioblastoma cells, which leads to disruption of the normal course of apoptosis in them. The review thoroughly highlights the latest information on both post-transcriptional modifications of certain regulatory RNAs, associated with the process of apoptosis, presents data on the features of apoptosis in glioblastoma cells, and shows the relationship between regulatory RNAs and the apoptosis in tumor cells. In conclusion, potential target candidates are presented that are necessary for the development of new drugs for the treatment of glioblastoma.

## 1. Introduction

Over 150 different variants of RNA modification have been described in databases [1,2,3]. These modifications have been shown to be essential for the post-transcriptional life of RNA; as with many other reversible reactions, this turns out to be a subtle tool for the wide and flexible control of the processes of cellular life. The most common modifications of RNAs are: 2′-O-methylation (Nm), pseudouridylation (Ψ), N6-methyladenosine (m6A), 5-methylcytosine (m5C), N1-methyladenosine (m1A), N7-methylguanosine (m7G), editing of adenosine (A) to inosine (I) or cytidine (C) to uridine (U) [4]. The m6A/m5C/m1A/m7G modifications are dynamically regulated by the proteins: writers, erasers and readers, which predominantly recognize sites of modification that affect the fate of RNA [5]. Human transfer RNAs (tRNAs) also undergo modifications including various base methylations, Ψ, Nm, etc., which is necessary to maintain tRNA stability, modulate tRNA folding and to accurately and efficiently decode [6]. Methylation protects the 3’-UTR of mRNA from hydrolysis, prolonging its lifetime [7]. Moreover, methylation and/or other modifications can stabilize the 3’-endo-conformation, provide an additional opportunity for the formation of hydrogen bonds, in particular to improve interaction with ribosomal proteins and other RNA-binding proteins and thereby affect the translation status [8]. Changes in the level of proteins that regulate the RNA modifications affect changes in both the expressed transcript repertoire itself and the cellular proteome and phenotype.

There are more and more studies of the role of post-transcriptomic modifications in the development and maintenance of carcinogenesis, in particular glioblastoma. Glioblastoma is one of the aggressive brain tumors, with an average patient life expectancy of about 15 months [9]. Attempts to develop diagnostics and a targeted approach in the treatment of glioblastoma continue. Due to the fact that post-transcriptome modifications of RNA are vital for the cell, it is of interest to study their role in human glioblastoma cells in order to develop both diagnostic signatures and promising targets for targeted therapy [10]. Previously, successful attempts have been made to thoroughly describe the role of RNA modifications in glioblastoma [10,11], including glioblastoma stem cells (GSCs) [12]. However, a comprehensive review of this topic, taking into account the role of non-coding RNAs (ncRNAs) and their possible modifications in the development of oncogenesis, is currently lacking. Therefore, we have included subsections on post-transcriptional modifications of microRNAs, subsections on small nucleolar RNAs (snoRNAs) and long non-coding RNAs (lncRNAs), which are directly or indirectly involved in RNA modifications and thus affect tumorigenesis. Moreover, we tried to focus not on a comprehensive description of all possible RNA modifications but on the subtle relationship between post-transcriptional modifications in RNA and the negative regulation of apoptosis in glioblastoma. 

In particular, this review attempts to summarize the knowledge accumulated in recent research about the role of modifications in the development of glioblastoma: the activation of its signaling pathways, the designation of key links in these pathways, and the sometimes dual, ambiguous role of certain modifier proteins. An analysis of the available literature showed that the regulators of RNA modifications interact with each other. Key methylation proteins (writers, readers, erasers) can interact with lncRNAs [13]. Those, in turn, can act quite independently, binding directly to their mRNA, DNA or protein targets. It should be noted that, during splicing of lncRNAs, small nucleolar RNAs are formed, which in turn direct RNA methylation in the 2-O position, isomerize uridine to pseudouridine, and modify ac4C of 18 S rRNA [14]. It was found that RNA regulators can modulate the process of apoptosis, so we have devoted separate sections to the features of this process in human glioblastoma cells and to the relationship between RNA modifications and the apoptosis. Many studies have shown a connection with one or another regulator and pro-apoptotic or anti-apoptotic proteins [15]. Thus, by modulating the level of specific regulators, it is possible to activate the processes of apoptosis in tumor cells which are necessary for inhibiting glioblastoma cells. In conclusion, we present the main axes and their regulators, which may be useful for the development of prognosis-predictive signatures or for the development of targeted anti-cancer therapies aimed at activating apoptotic signaling pathways in glioblastoma.

## 2. Post-Transcriptional Modifications

### 2.1. N6-Methyladenosine (m6A)

One of the most studied and most common mRNA modifications is N6-methyladenosine (m6A). A popular strategy used in many studies to establish the role of m6A methylation in biological processes is to suppress the expression level of m6A-related proteins and elucidate the effects. Because m6A methylation is a vital process, it encompasses many cellular events, including splicing, translation, nuclear export and degradation. There are three groups of proteins whose function is associated with m6A-methylation: m6A methyltransferases (writers), demethylases (erasers) and m6A binding proteins (readers). Interaction of these proteins with RNA results in changes in RNA stability and functionality.

#### 2.1.1. Writers

Methylation of adenosine in N6-position is carried out by a protein complex consisting of several subunits. The core components of the complex are methyltransferase 3 (METTL3) and methyltransferase 14 (METTL14) proteins. METTL3 is a key protein because of its catalytic activity. METTL14 acts as an RNA-binding protein without catalytic activity. Auxiliary proteins such as WTAP, KIAA1429, HAKAI, RBM15 are required for proper positioning and functioning of the complex. There are often no consistent data on the effects of m6A levels on in vitro and in vivo processes. Silencing the m6A methyltransferase leads to altered gene expression and alternative splicing patterns, resulting in activation of the p53 signaling pathway and apoptosis [16]. A decreased level of m6A stimulates epithelial–mesenchymal transition (EMT) and vasculogenic mimicry (VM) in glioblastoma cells. Knockdown of METTL3 could promote the EMT and VM process by regulating MMP2, CDH1, CDH2 and fibronectin 1 [17]. 

METTL3 expression is dramatically important in GSCs. Modulation of m6A-related protein level may be a key to the elucidation of their role and the role of m6A in GSCs’ response to various types of treatment. Overexpression of METTL3 maintains stemness of GSCs largely through the upregulation of SOX2 level by targeting 3′-UTR of SOX2 mRNA. Knockdown of METTL3 enhanced sensitivity to γ-irradiation and reduced DNA repair inhibit neurosphere formation and maintain GSCs [18]. Furthermore, METTL3 is a potential target in temozolomide-resistant glioblastoma. It is shown that TMZ increased the level of METTL3, which enhances the methylation of histone modifiers. TMZ-resistant cells with knockdown of METTL3 have been shown to reduce orthotopic TMZ-resistant xenograft growth in vivo. Authors have reported the SOX4/EZH2/METTL3 axis as a potential target in TMZ-resistant GBM [19].

#### 2.1.2. Erasers

The demethylase group of proteins currently consists of two enzymes: fat mass and obesity-associated protein (FTO) and α-ketoglutarate-dependent dioxygenase alkB homolog 5 (ALKBH5). FTO has more functions; this protein is active not only against m6A but also against m1A and m6Am, and also it demethylates nucleotides in both RNA and DNA. In addition, it was shown that FTO demethylates m6Am at the first position after the 7-methylguanosine cap structure and destabilizes transcripts in this way [20]. There are currently no consistent data on the oncogenic or tumor suppressor role of FTO in glioblastoma. So, it was shown that FTO knockdown has the opposite effect in in vitro and in vivo experiments: inhibited GSC growth and self-renewal in vitro inhibits tumor progression and increases the lifespan of GSC-grafted mice in vivo [21]. Furthermore, it was shown that FTO knockdown promotes m6A-mediated maturation of miR-10a leading to enhanced glioblastoma progression by targeting 3′-UTR of Myotubularin Related Protein 3 (MTMR3) mRNA and modulation of the Wnt/β-catenin pathway [22]. Pro-apoptotic proteins in this pathway (for example, cleaved caspase-3) are downregulated, and antiapoptotic (Bcl-2, c-myc, and p-c-Jun) proteins are upregulated by an miR-10a-mediated mechanism. Decreasing the FTO transcriptional activity may be carried out by transcription factors (TF), for example, SPI1 [23], which is upregulated in glioblastoma cells and has antiapoptotic activity [24]. In contrast, there are studies that show FTO as an oncogene [25]. FTO upregulated the MYC level, known as proto-oncogene TF [26], and downregulated the expression of MAX interactor 1 (MXI1), which binds to the MYC promoter and inhibits its expression, playing an oncosuppressor role in glioblastoma [27,28].

ALKBH5 exhibits activity only against m6A but not m6Am or m1A, in contrast with FTO. The nuclear localization of ALKBH5 [29] indicates that this protein acts on immature forms of mRNA as well as on non-coding RNAs widely present in the nucleus: long non-coding RNAs (lncRNAs), small nuclear (snRNAs) and small nucleolar RNAs (snoRNA) [30]. RNA demethylase ALKBH5 is highly expressed in glioblastoma stem cells (GSCs) and promotes tumorigenicity through the binding mRNA of the FOXM1 transcription factor by interacting with lncRNA FOXM1-AS. ALKBH5 demethylates nascent FOXM1 transcripts, which leads to enhanced FOXM1 expression in GSCs [31].

#### 2.1.3. Readers and Other m6A-Related Proteins

Bioinformatic studies have shown that 12 m6A-related genes, including m6A-readers (YTHDC1, YTHDC2, YTHDF2, HNRNPC, LRPPRC, HNRNPA2B1, IGFBP1, IGFBP2, IGFBP3, RBMX), may play a crucial role in glioma progression. These genes are involved in Myc- and Hedgehog signaling pathways leading to a decreasing survival rate in glioma [32]. YTH N6-Methyladenosine RNA Binding Protein 2 (YTHDF2) knockout decreased the expression of MYC targets and many transcripts involved in essential processes: SOX2 and OLIG2 (stemness markers), VEGFA, IGF-binding proteins, ribosomal proteins. Moreover, overexpression of YTHDF2 results in activation of the YTHDF2-MYC-IGFBP3 axis in in vitro and in vivo tumor growth [33]. 

Some proteins may regulate m6A methylation by attracting the methyltransferase complex to near their own target site without being a classic m6A-related protein. For example, serine/arginine-rich splicing factor 7 (SRSF7) recruits the methyltransferase complex and facilitates the m6A methylation near its binding sites. This methylation upregulates the proliferation and migration of glioma cells [34].

### 2.2. Other RNA Modifications

#### 2.2.1. Editing of Adenosine (A) to Inosine (I) 

Deamination is adenosine-to-inosine base editing, which strongly modulates the expression of target RNAs. Adenosine deaminases acting on the RNA (ADAR) protein family consist of three members: ADAR1, ADAR2, and ADAR3. ADAR1 has two promoter-specific splice variants of their transcript: the p110 isoform has mainly nuclear localization, and p150 can be found in the nucleus and in cytoplasm. Acting as a homodimer, ADAR1 binds dsRNA regions. For example, ADAR1 controls antiapoptotic genes expression by editing 3′-UTR in double-stranded RNA, negatively affecting the expression of their protein products. The most ADAR1-edited protooncogenic mRNAs are mRNAs of XIAP and MDM2, which participate in the regulation of apoptosis in tumor cells. It is interesting that ADAR1 contends with the STAU1 (RNA shuttling factor) to contribute the nuclear retention of MDM2 and XIAP transcripts, apparently to increase the deamination of XIAP mRNA. Decreasing ADAR1 expression or catalytic activity promotes XIAP-mediated apoptosis suppression, and the ectopic expression of ADAR1 causes cell death [35].

At the same time, there are interactions between various proteins performing RNA modifications. It is very interesting that METTL3, which is upregulated in glioblastoma cells, targets ADAR1 mRNA to stabilize their transcript and increase ADAR1 protein level. Due to the upregulation of METTL3 in glioblastoma, a high level of m6A complicates the deaminase activity of ADAR1. In these conditions, ADAR1 acts as an RNA binding protein without modification potency, for example, binding CDK2 to promote their stability. CDK2 is a crucial regulator of cell cycle [36], and a high level of CDK2 boosts the proliferation of cancer cells including glioblastoma [37]. Thus, there is an important new interaction—METTL3/ADAR1/CDK2—as a potential target for glioblastoma therapy [38].

#### 2.2.2. N1-Methyladenosine (m1A) RNA Modification

N1-methyladenosine (m1A) RNA modification is a frequent modification of non-coding RNAs—tRNA, rRNA, lncRNA—which is rarely encountered for mRNA. Writers, erasers and readers for m1A modification have been identified. Writers of m1A contain tRNA Methyltransferase 6 Non-Catalytic Subunit (TRMT6)—which can be located in the nucleus or cytoplasm and which methylates adenosine in tRNA as well as in mRNA [39,40]—and nucleomethylin, which has only nucleolar localization and methylates 28S rRNA [41,42]. Erasers of m1A include ALKBH1, ALKBH3 and FTO [43,44,45]. Readers of m1A comprise YTHDF1-3 and YTHDC1 [46,47]. TRMT6 plays an oncogenic role in glioblastoma cells and patients: it was shown that TRMT6 is upregulated in glioblastoma, and their knockdown via siRNA decreased glioma cells’ invasion, migration and proliferation. Bioinformatic analysis has shown that TRMT6 potentially influences glioma progression by being involved in cell cycle regulation, and the upregulation of PI3K-AKT, TGF-beta, mTORC1, NOTCH and MYC signaling pathways and other vital processes [48]. Other bioinformatics assays revealed that a high level of m1A “writer” YTHDF2 is associated with the WHO grading of gliomas and is associated with poor prognosis [49]. 

#### 2.2.3. 5-Methylcytosine RNA Modification

5-methylcytosine (m5C) modification plays a significant role in cellular events including carcinogenesis. There are erasers, writers and readers for m5C methylation similar to m6A, but m5C-related processes are less studied. Several proteins constitute m5C writers: NOL1/NOP2/SUN domain (NSUN) family (NSUN1-7), tRNA-specific methyltransferase (TRDMT) family members, DNA (cytosine-5-)-methyltransferase 2 (DNMT2) [50,51,52]. Erasers of m5C consist of the TET proteins family and ALKBH1 protein. It is interesting that m5C erasers have multiple functions not only in RNA, but also in DNA modifications. TET family proteins catalyze 5-methylcytosine (5mC) to 5-hydroxymethylcytosine (5hmC) in DNA. TET1 regulates 5-formylcytosine (5fC) to 5-carboxylcytosine (5caC) oxidation. TET2 carried out RNA 5hmC modification. ALKBH1 catalyzes m1A demethylation in cytoplasmic tRNA and other tRNA modifications. Currently, readers of m5C contain two proteins: Aly/REF export factor (ALYREF) and Y-Box Binding Protein 1 (YBX1) [50]. ALYREF recognizes m5C in RNA located in the nucleus [53] and YBX1 binds cytoplasmic RNA [54]. Furthermore, a recent study has shown that the Fragile X mental retardation protein, the mutation that causes Fragile X syndrome, may play an m5C reader role [55].

In glioma cells, not only RNA but also DNA modifications are aberrant [56]. One of the most common modifications of DNA is the methylation of cytosine in specific sites, called CpG islands [57]. 5′-CpG islands in the promoter region of NSUN5, a writer of m5C, are methylated in glioblastoma. Epigenetic silencing of NSUN5 promotes dysregulation of m5C modification, one of the most significant changes of which is an unmethylated cytosine in the 3782 position of 28S rRNA. This status leads to the overall downregulation of protein synthesis due to influence on stability of the rRNA-tRNA-mRNA complex, disturbing the structure of the P-site at the edge of the 28S subunit. Silencing NSUN5 distorts translation and downregulates protein synthesis by affecting not only rRNA but also tRNA. Interestingly, the epigenetic downregulation of NSUN5 is potentially an adaptive mechanism to fight stress conditions by reducing high energy-consuming processes such as protein synthesis. In general, patients with methylated promoter CpG islands of NSUN5 have a better prognosis [58]. Bioinformatic analysis of the expression of the m5C-related methyltransferases NSUN protein family revealed an association between its altered expression and the malignant progression of glioblastoma [59].

#### 2.2.4. 7-Methylguanosine (m7G) RNA Modification

The m7G modification protects and stabilizes transcripts from exonucleolytic degradation and influences the processing of the mRNA molecules. In eukaryotes 7-methylguanosine (m7G), tRNA modification is catalyzed by the methyltransferase complex formed by METTL1 and WDR4 [60]. It was shown that the suppression of METTL1 or WDR4 in tumor cell lines in in vitro experiments leads to the arrest of cell proliferation, increased apoptosis, reduced colony formation and migration, and reduced tumor formation in in vivo experiments [61]. Recently, the effect of m7G RNA methylation regulators on glioma prognosis has been investigated, where three genes (NUDT7, NUDT11 and CYFIP2) have been proposed as promising prognostic biomarkers for gliomas [62]. It is worth noting that Cytoplasmic FMR Interacting Protein 2 (CYFIP2) is a direct p53 target gene responsible for p53-dependent apoptosis [63].

## 3. Non-Coding RNA and Apoptosis

### 3.1. Effects of Long Non-Coding RNA on Apoptosis

Long non-coding RNAs (LncRNAs) are known to play a key role in oncogenesis and are currently being studied in detail for the development of new targeted oncolytic drugs. LncRNAs are about >200 nucleotides; their functional role has been extensively investigated. The functional versatility of lncRNAs is related to their ability to conform to various structures and molecular interactions with DNA, RNA and proteins [64]. The relationship of lncRNAs to the development of glioblastoma has already been described in a number of reviews [65,66,67] and studied in meta-analyses [68,69], which have already identified potential targets for developing diagnosis, prognosis assessment, and combination therapy. LncRNAs can indirectly affect the post-translational modifications of RNA, and hence can affect processes of oncogenesis such as apoptosis. For example, lncRNA—just proximal to the X-inactive specific transcript (JPX)—mediated apoptosis in glioblastoma cells in an m6A-dependent manner by promoting FTO/PDK1 interaction [70]. PDK1 is one of the important proteins in the PI3K-Akt signaling pathway; the last, when activated, leads to the inhibition of the apoptosis process in the cell through mediators such as MDM2, IκB and BAD [71].

The lncRNA Sox2 overlapping transcript (SOX2OT) is known to be upregulated in various types of cancers, regulating cell proliferation, cycle arrest, apoptosis, migration, invasion and metastasis. The gene of SOX2OT contains within its intron the SRY-box transcription factor 2 gene—SOX2—transcribed in the same orientation. Recently, it was shown that SOX2OT increased SOX2 expression and thus activated the Wnt5a/β-catenin signaling pathway which increased expressions of downstream target genes, such as CyclinD1, C-myc, LEF1, TCF1/7, Met/pro-Met [72]. SOX2OT also regulates the level of SOX2 transcription factor, which plays a key role in in maintaining the pluripotency of cancer stem cells. In addition, SOX2OT itself may promote cancer progression through sponging miRNA-144-3p, which induces cell apoptosis and targets the c-MET oncogene. Several studies showed a connection between SOX2OT and the activation of the PI3K/Akt signaling pathway through the sponging of definite miRNA [73,74]. Unfortunately, the SOX2OT–SOX2 interaction results in increasing the SOX2 level, leading to higher TMZ resistance of glioblastoma. Moreover, an enhanced level of SOX2 protein is achieved by recruiting RNA demethylase ALKBH5 by SOX2OT and subsequent demethylation of the SOX2 transcript to stabilize SOX2 mRNA [72]. Therefore, SOX2OT inhibited apoptosis, and increased the proliferation and TMZ resistance of glioblastoma in m6A-dependent manner. The authors concluded that LncRNA SOX2OT is suitable as a new biomarker for the prognosis of glioblastoma and as a therapeutic target for reducing TMZ chemoresistance. 

Metastasis associated lung adenocarcinoma transcript 1 (MALAT1) is one of the target lncRNAs for METTL3. Methylation of MALAT1 by METTL3 increases its stability. Upregulated MALAT1 activated transcriptional factor NF-kB. Interestingly, the RNA-binding protein HuR, which is reported to have antiapoptotic activity [75,76], is essential for MALAT1 modification and MALAT1-mediated NF-kB activation. Moreover, it plays an important role in m6A methylation of SOX2 mRNA, leading to its activation [77]. 

A recent study [78] using clustered regularly interspaced short palindromic repeats interference (CRISPRi), based on the stable expression of the dCas9-KRAB complex, revealed novel potential lncRNA targets for radiotherapy-resistant glioblastoma therapy. Thus, 5689 lncRNA loci were screened and nine lncRNAs—called lncRNA Glioma Radiation Sensitizers (lncGRS)—were the most associated with glioblastoma cell growth in the presence of radiation. LncGRS-1, localized in the nucleus and conserved among primates, generated the most interest. It was shown that ASO-mediated knockdown of lncGRS-1 selectively inhibited growth rate and increased the radiation sensitivity of tumor cells but not normal cells. This makes lncGRS-1 a very attractive potential target for glioblastoma targeting therapy.

Recently, a signature containing 12 m6A/m5C/m1A/m7G-related lncRNA genes (AL080276.2, AC092111.1, SOX21-AS1 [79], DNAJC9-AS1, AC025171.1, AL356019.2, AC017104.1, AC099850.3, UNC5B-AS1, AC006064.2, AC010319.4, and AC016822.1) was investigated to predict the prognosis of glioblastoma patients [80]. This study revealed new targets for glioblastoma therapy and future research directions. 

### 3.2. Effects of Small Nucleolar RNA on Oncogenesis

Recently, it was shown that a subset of ncRNAs, snoRNAs, participated in tumorigenesis [81]. In general, snoRNAs are approximately 60–300 nucleotides in length, predominantly accumulate in nuclei and assist ribosomal RNA (rRNAs) in their catalytic modifications. Based on structure, snoRNAs can be divided into two groups: C/D-box snoRNAs (SNORD), which control the 2′-O-methylation of rRNAs, and H/ACA-box snoRNAs (SNORA), which control the pseudouridylation of rRNAs [82]. Interestingly, some proteins that form ribonucleoprotein complexes with SNORA and perform pseudouridylation in rRNA can also mediate apoptosis. For example, DKC1, which is elevated in glioblastoma, correlates with WHO grade and a poor prognosis of glioblastoma. In addition, DKC1 suppresses CDK2 and cyclin E2 levels, leading to the dysregulation of the cell cycle [83].

Depending on the type of cancer, snoRNAs play different roles, both tumor suppressor and oncogenic. In various literature sources, one can find the relationship between the level of snoRNA and processes such as metastasis, invasion and angiogenesis, etc. [84,85,86,87]. Studies related to the role of SNORD in the progression of glioma are primarily associated with SNORD44 [88], SNORD47 [82] and SNORD76 [89].

In small nucleolar RNA, SNORD44 was downregulated in breast cancer, colorectal cancer and the head and neck squamous carcinoma, which correlated with the prognosis for the patient [90,91,92]. In the glioma, it was also shown that the levels of the SNORD44 and the Growth Arrest Specific transcript 5 gene (GAS5) were abnormally downregulated in a significant positive correlation in the glioma tissues and cells, which also correlated with the WHO grade. Co-overexpression of SNORD44 and GAS5 due to transfection suppressed invasion, migration, cell proliferation and led to apoptosis; the latter is confirmed by the increase of the expressions of cleaved PARP, caspases 3, 8 and 9 [88]. Similar results were obtained on the colorectal cancer cells, where the co-overexpression of SNORD44 and GAS5 leads to the caspase-dependent apoptosis, resulting in the inhibition of the cell proliferation [91]. 

SNORD47 plays a similar role in the pathogenesis of gliomas. It has also been shown that its level is reduced in human glioblastoma tissues and is also inversely correlated with the WHO grade [93]. Overexpression of SNORD47 results in the decreased expression of β-catenin and p-β-catenin. These proteins are central to the Wnt signaling pathway, which can inhibit the apoptosis and promote cell proliferation [94]. SNORD47 also inhibited the epithelial mesenchymal transduction by suppressing nuclear translocation and β-catenin activation [95]. SNORD47 demonstrated a synergistic effect in combination with temozolomide and enhanced glioma temozolomide sensitivity.

SnoRNA that is excised from the third intron of GAS5, SNORD 76, is being considered as a potential tumor suppressor in GBM [89]. It was shown that lower SNORD76 expression correlated with WHO grades III and IV of patients and poorer overall survival. At the cellular level, tumor SNORD76 suppression may be associated with changes in the levels of cyclin D1 and p21 protein. Earlier, it was found that the expression of SNORD76 leads to retinoblastoma gene (Rb)-associated cell cycle arrest in S phase, inhibiting the tumorigenicity of glioma cells [89]. Therefore, these SNORDs (44,46,76) could be potential prognostic indicators and therapeutic targets for glioblastoma. Meanwhile, more research is needed to identify the exact mechanisms involved in this regulation, in which RNAs undergo 2-O-methylation by SNORD44, SNORD 46 and SNORD 76; is it the epigenetic modulation or the repression of transcriptional regulation?

### 3.3. Effects of Post-Transcriptional Modifications of microRNAs on the Oncogenesis

MicroRNAs (miRNAs) are short non-coding RNAs 19–24 nt in size that, through translational repression or mRNA degradation, affect gene expression at the post-transcriptional level. For this reason, miRNAs are involved in the regulation of various cellular processes by destroying their intracellular mRNA. Aberrant expression of miRNAs can influence the emergence, progression and metastasis of tumors, including glioblastomas [96]. Quite a lot of research is devoted to the search for diagnostic signatures of miRNAs and their levels in the context of the development and further prognosis of glioblastoma [97,98].

MicroRNAs themselves can undergo post-transcriptional modification, representing an additional layer of microRNA function regulation in cancer. The first evidence of post-transcriptional modifications of miRNAs was obtained in 2004 [99], when the A-to-I conversion within the miR-22 [100] precursor was shown. Since then, the role of other miRNA post-transcriptional modifications in the promotion and progression of cancer, such as m6A, m7G and m5C, has begun to be elucidated [101]. With regard to glioblastoma, more and more data are emerging about specific microRNA modifications affecting progression, invasion, apoptosis, and other processes of oncogenesis. 

METTL3 and METTL14 are two of the main «writers» for the m6A modification of miRNAs. It was shown earlier that METTL3-mediated m6A RNA modification is critical for glioblastoma stem cell maintenance and dedifferentiation of glioma cells [18]. The overexpression of METTL14 also leads to decreasing ASS1 expression, promoting cell proliferation, migration, and invasion in glioma [102]. At the same time, the knockdown of METTL3 and METTL14 dramatically promotes human glioblastoma stem cell growth, self-renewal, and tumorigenesis [21]. The dual action of METTL14 as a tumor suppressor and tumor promoter is discussed in detail in reference [103]. 

The inhibition of mature miRNA biogenesis or its dysregulation as a result of the A-to-I editing of miRNA precursors leads to the aggressive growth of glioma (in the case of miR-376a-5p [104]), the proliferation and migration of glioblastoma cells (miR-221/222 and miR-21), and cell invasion (miR-589-3p) [101]. 

The relationship between post-transcriptional modifications of specific microRNAs and apoptosis has been investigated so far only in a few studies. The m6A modification prevents pri-miR-125b-2 processing into mature miR-125b; the latter regulates cell growth and invasion in pediatric low grade glioma, resulting in decreased growth and invasion, as well as in induction apoptosis [105]. The overexpression of METTL3 in different types of cancer promotes the m6A modification of pri-miRNAs, which leads to the upregulation of mature oncomirs: miR-1246, miR-25-3p, miR-221/222, miRNA-92 and miR-126-5p. These oncomirs reduce the level of PTEN or/and activate the PI3K/Akt/mTOR pathway, both of which play important roles in the regulation of apoptosis [71,101]. In glioblastoma, m5C modifications of mature miR-181a-5p lead to the loss of its ability to target the mRNA of the pro-apoptotic protein BIM, also known as B-cell chronic lymphocytic leukemia/lymphoma (Bcl-2)-like 11 (BCL2L11) [106]. The enzyme responsible for m7G modification, METTL1, is often overexpressed in glioma and is associated with the poor prognosis of patients [107]. Several studies have shown that METTL1 knockdown significantly increased cell apoptosis [108,109].

The situation is complicated by the fact that miRNA binding sites themselves can be modified by m6A or A-to-I editing, which positively or negatively affects miRNA–mRNA pairing through several different mechanisms. This has been found for the VEGFA, SEPT2 and IGF1R oncogenes, which can be rescued from miRNA inhibition by m6A modification in the 3’UTR. It was previously reported that miRNA-15b and miRNA-29b, which were identified in this analysis, inhibit stemness in glioma cells and induce apoptosis [110].

The study of functional activity, the level of modified microRNAs or signatures of modifications of the 3′-UTR miRNA binding region, and of samples of primary cell lines or samples of patients with different WHO grades of glioma, is necessary in the future.

Table 1 summarizes all the non-coding RNAs we mentioned in the review and their roles in glioblastoma cells.

## 4. The Main Regulators of Apoptosis in Glioblastoma Cells

One of the key features or hallmarks acquired by a tumor cell is the ability to evade programmed cell death, particularly apoptosis. There are three pathways for the initiation of apoptosis: (1) extrinsic or death receptor mediated pathway, which involves trans-membrane receptor mediated interactions; (2) intrinsic pathway which starts through the mitochondrial-initiated events described; (3) alternate apoptotic pathway initiated by either perforin and granzymes (A, B) or via phagocytosis directly by cytotoxic T cells. Genes that play a key role in the development of apoptosis are divided into pro-apoptotic (genes of caspases, tumor necrosis factor (TNF) genes (ligands and receptors), Bax, Bak, Bclx, Bad, Hrk, Bid, Bik, Blk) and anti-apoptotic (Bcl-2, Bcl-x, Bcl-XL, Bcl-XS, Bcl-w, BAG, MCL-1, A1/BFL-1, IAP, TRAF, AKT1, BRAF and BFAR) genes. The main morphological, molecular and biochemical changes that occur during apoptosis (cell shrinkage and pyknosis) can be identified using microscopy, flow cytometry and using different biochemical methods for the measurement of caspase activity, DNA fragmentation, analysis of chromatin changes, methods to measure membrane potential and permeability transition in the mitochondrial DNA, analysis of the mitogen-activated protein kinase (MAPK) pathway, cell cycle regulatory kinase Cdk2, and the measurement of changes in intracellular calcium activity, etc. 

In cancer, the normal mechanisms of the regulation of apoptosis do not function; either hyperproliferation of cells or a decrease in cell deletion are observed. Moreover, the suppression of apoptosis leads to tumor initiation, progression or metastasis [111,112]. There are many ways in which a tumor cell can reduce apoptosis or be resistant to apoptosis, namely: (1) impaired receptor signaling pathway due to reduced expressions of death receptors or death signals; (2) disrupted balance of the B-Cell Leukemia/Lymphoma 2 (Bcl-2) family of proteins, particularly overexpression anti-apoptotic proteins (Bcl-2) or down-regulation or mutation of pro-apoptotic proteins (BCL2 Associated X (Bax)); (3) reduced expression caspases; (4) increased expression of the inhibitors of apoptosis proteins (IAPs); and (5) defects or mutations in the p53 gene or its negative regulation [113,114,115]. 

In the process of studying the features of apoptosis in glioblastoma cells, it was found that all of the abovementioned strategies are used to escape tumor cells from cell death [71]. Great apoptosis deregulation in glioblastoma cells was demonstrated. Thus, the development of drugs aimed at activating apoptosis in glioblastoma cells is a promising strategy. The evasion of apoptosis by glioblastoma cells could be explained by the extrinsic death receptor pathway that is related to Fas/FasL signal transduction being inhibited, especially at the stage of caspase-8 activation [116]. It was also shown that one of the members of the TNF family of proteins—the tumor necrosis factor-related, apoptosis-inducing ligand (TRAIL)—induces apoptosis in glioblastoma cells through binding to its death receptors, DR4 and DR5 [117,118,119]. In glioblastoma cells, the overexpression of anti-apoptotic proteins (Bcl-2) was observed. This protein inhibited the activation of Bax and Bad, which led to the permeabilization of the mitochondrial outer membrane (MOM) and the release of cytochrome c from the mitochondria [120]. Respectively, Bcl-2 inhibits the formation of the apoptosome, the activation of caspases, and eventually cell death [121]. It was shown that, in the glioblastoma anti-apoptotic proteins, Bcl-2 is overexpressed. Another strategy is the downregulation of apoptosis-stimulating genes; statistically significant changes of expressions, not only BAX but also NLRC4, CASP10, DAP1 and BIRC5 genes, were demonstrated [122]. The observed lower protein levels of caspase-3 and procaspase-9 leads to the inhibition of the intrinsic mitochondrial apoptotic pathway [123,124]. Moreover, METTL3 deficiency decreases m6A modification, leading to the downregulation of pro-apoptotic protein, BCL-X isoform (e.g., BH3-only proteins, including BIM, PUMA, and BAD) [125]. Inhibitors of apoptosis proteins (IAPs) block the activity of caspases 3, 7 and 9 by binding to their catalytically active pockets, representing the final molecular barrier to successful apoptosis [126]. In glioblastoma, the overexpression of various IAPs has been observed, which correlated with poor clinical outcomes in patients. A study of apoptosis in glioblastoma cells showed a differential level of proteins involved in the PI3/Akt/mTOR intracellular signaling pathway. In GBM, phosphorylated RAC-alpha serine/threonine-protein kinase (AKT) induces the overexpression of MDM2 protooncogene, an important negative regulator of p53, and inhibits the apoptosis-inducing protein, Bad, by its phosphorylation [71,127]. Moreover, AKT-phosphorylation of IκB promotes the nuclear translocation of NFκB and the subsequent regulation of its target genes, including IAPs [128]. Other players also intervene in such a complex regulation of apoptosis. For example, a tumor suppressor gene, phosphatase and tensin homolog, deleted on chromosome 10 (PTEN) and functioning as a lipid phosphatase, prevents the activation of AKT, thereby blocking the PI3K/AKT/mTOR pathway [129]. Unfortunately, PTEN is inactivated in glioblastoma cells as on the genetic level (gene mutations, gene deletions) and on the epigenetic level (methylation DNA, low level of mRNA) [130]. Figure 1 shows the main features of the course of apoptosis in glioblastoma.

Recently, special attention has been paid to the relationship between the development of apoptosis and the epigenetic regulation of gene expression (methylation and histone modifications), post-transcriptional regulation (RNA modifications) and post-translation modification of proteins [131,132,133,134]. 

## 5. The Relationship between RNA Modifications and the Regulation of Apoptosis

Summarizing the above, we would like to draw attention to the following key links associated with post-transcriptional modifications and the development of oncogenesis, in particular apoptosis, in terms of the possible development of both diagnostic signatures and the development of drugs for targeted cancer therapy. Among the post-transcriptional modifications, the main role remains for m6A; that is why m6A-related proteins can be considered for the development of predictive signatures or for the development of targeted anticancer therapies aimed at activating apoptosis in glioblastoma cells.

One of the main proteins in m6A, METTL3, is known to regulate apoptosis by targeting Bcl-2 in different types of cancer [135,136]. Knockdown of METTL3 induces apoptosis in lung cancer cells by affecting apoptosis-related proteins (Bax, Bcl-2, PARP, Caspase 3) and inhibits the phosphorylation of AKT leading to the suppression of the PI3K/Akt pathway [137]. METTL3 siRNA-mediated downregulation decreases mRNA and protein levels of Neurogenic locus notch homolog protein 3 (NOTCH3) [138], Delta-like ligand 3 (DLL3) [139] and Hairy and Enhancer of Split-1 (HES1) [140]. Knockdown of METTL3 inhibits the proliferation of glioblastoma cell lines. These data show that m6A methylation influences the NOTCH pathway, which is a potential target for glioblastoma treatment [141]. It is known that the NOTCH pathway influences many vital cell processes including apoptosis, so activation of this pathway by m6A-mediated stabilization of NOTCH3, DLL3 and HES1 mRNA, and increased expression of corresponding proteins, leads to the inhibition of apoptosis. Methylated RNA immunoprecipitation-seq (MeRIP-seq) and integrated transcriptome analyses showed that decreasing the expression of METTL3 downregulated m6A modification levels of serine and arginine-rich splicing factors (SRSF), which promoted the YTHDC1-dependent non-sense mediated decay of SRSF transcripts and decreased SRSF protein expression, and that is why apoptosis was activated [142]. 

One of the main erasers of m6A, ALKBH5, which is upregulated in glioblastoma cells, demethylated glucose-6-phosphate dehydrogenase (G6PD) mRNA, promoting the translation and activation of the pentose phosphate pathway (PPP) [143]. Knockdown of G6PD enhances apoptosis [144], so ALKBH5 could mediate apoptosis in a G6PD-dependent manner.

The m6A-reader, YTHDC1, inhibits the protein level of Vacuolar protein-sorting-associated protein 25 (VPS25). This protein belongs to the endosomal sorting complex required for transport [145]. In glioblastoma cells, VPS25 mediated the phosphorylation of Janus kinase (JAK) and signal transducers and activators of transcription (STAT) proteins, and activates this pathway. However, VPS25 inhibits apoptosis in glioblastoma [146]. Pre-mRNA-splicing regulator WTAP promotes the m6A-methylation of the Heat Shock Protein Family A (Hsp70) Member 7 (HSPA7) pseudogene in nuclear speckles. HSPA7 overexpression in glioblastoma activates various carcinogenic pathways including apoptosis [147].

Other promising targets in terms of developing diagnostic signatures and targeted drugs are non-coding RNAs (lncRNA and snoRNAs). Undoubtedly, it is lncRNA SOX2OT, which is proposed as a new biomarker for the prognosis of glioblastoma and as a therapeutic target for reducing TMZ chemoresistance. Activation by SOX2OT’s two signaling pathways (Wnt5a/β-catenin and PI3K/AKT/ERK) leads to inhibiting apoptosis and cancer progression. The modulation of another lncRNA JPX can also be interesting in the development of a new targeted drug because of its promotion of FTO/PDK1 interaction and indirect inhibition of the apoptosis process. It will also be interesting to analyze the usefulness of the lncRNA gene signature associated with m6A/m5C/m1A/m7G in a larger sample of glioblastoma patients and with a functional analysis. 

Very promising are small nucleolar RNAs transcribed from GAS5, which also influence apoptosis and tumorigenesis. Although the exact mechanism is unclear, overexpression of SNORD44, SNORD47, and SNORD 76 is known to suppress cell proliferation and lead to apoptosis. The level of these SNORDs can also be useful as a prognostic indicator, since low levels of some SNORDs can be negatively correlated with WHO grades III and IV of patients and poorer overall survival. Figure 2 shows the structures of the major RNA modifications mentioned in the review and the changes in the levels of the most important effector molecules that regulate apoptosis.

## 6. Conclusions

In this review, we have highlighted recent data on the role of RNA modifications in the regulation of apoptosis in glioblastoma. Analysis of the literature shows that the dysregulation of RNA modifications in glioblastoma is an important change in cell life, powerfully influencing many cellular processes, including apoptosis. This can be either a direct effect by changing the methylation or deamination of the mRNA of genes encoding proteins associated with apoptosis, or indirect, by modifying ncRNAs that affect the activation of signaling pathways involved in apoptosis. In particular, we have identified the activation of signaling pathways such as PI3K/Akt, NOTCH, and JAK/STAT as the most important RNA modification-dependent cascades leading to changes in the rate of apoptosis in glioblastoma cells.

Due to the huge variety of both modifications and non-coding RNAs, approaches for diagnostics and for the development of targeted therapeutic agents are diverse. Recently, 12 lncRNA genes related to m6A/m5C/m1A/m7G have been proposed [80] that can accurately predict the prognosis of patients and set the direction for promising immunotherapy strategies in the future. For example, lncRNA SOX2OT [72,73,74] may be a promising target for glioblastoma therapy, including TMZ-resistant. The same is true for other lncRNAs mediating post-transcriptional changes in individual molecules and cell cycle regulation—JPX [70], lncGRS-1 [78] and FOXM1-AS [31]. However, in the case of SNORD44 [88,90,91], SNORD47 [82], and SNORD76 [89], as well as the lncRNA host gene GAS5 [15,91], a correlation was shown between their low expression and WHO grades III and IV of patients and poorer overall survival. Several studies have moved from exploratory to applied research into the search for diagnostic signatures of microRNAs, some of which mentioned above have already been published on ClinicalTrials.gov (miRNA-21, miRNA-25, miRNA-125, miRNA-221/222).

It is worth noting that the regulation of the level of modifications of specific target RNAs is a promising method for the development of glioblastoma therapy. Basically, the methods that are used as proof of principle are knockdown/knockout of the gene or silencing/interference of RNA; however, in order to develop an antitumor targeted drug, more detailed studies of the mechanisms of RNA–protein, RNA–RNA, and RNA–DNA interactions are necessary. Thanks to tools such as CRISPR/Cas, it is possible to influence protein levels by knockout of the protein-coding gene, as in the case of Cas9 [148]. The inhibition of target RNA levels was made possible by the discovery of CRISPR/Cas13 [149,150]. Proteins in this family carry out a specific targeted knockdown of the target RNA. Moreover, catalytically inactive Cas13 (dCas13) has strong and specific RNA-binding activity. This allowed the creation of fusion proteins (e.g., dRfxCas13d-ALKBH3 [45], dPspCas13b-ADAR2 [151], dRfxCas13d-ALKBH5 and dRfxCas13d-METTL3 [152]) for precise control of epitranscriptome modifications of individual target genes. Further development and research involving CRISPR/Cas13 is a promising direction for basic research on the role of individual ncRNAs, as well as a tool for targeted modification of the aberrant epitranscriptome profile of glioblastoma cells.

## Figures and Tables

**Figure 1 ijms-23-09272-f001:**
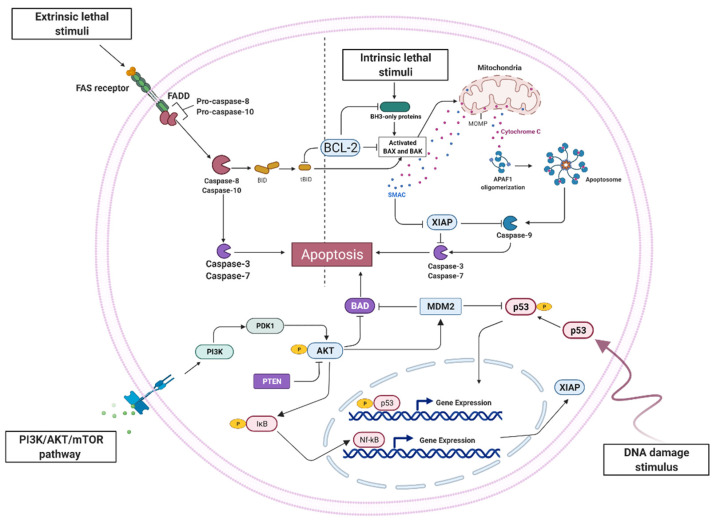
An overview of the signaling pathways that are involved in apoptosis in human glioblastoma cells.

**Figure 2 ijms-23-09272-f002:**
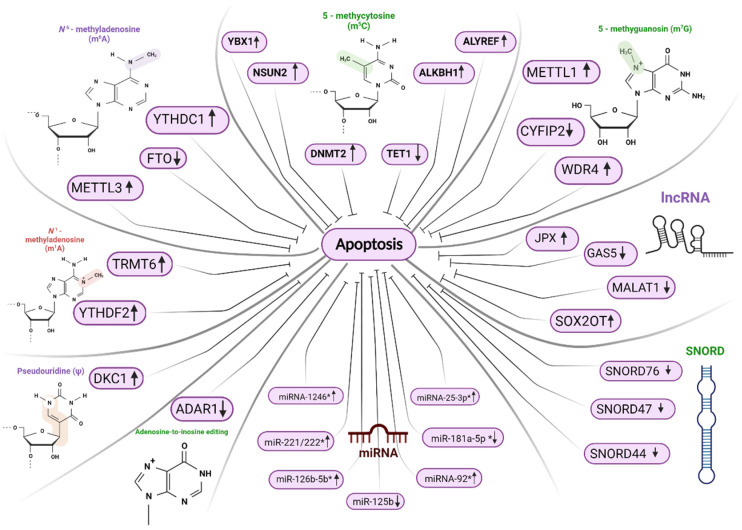
An overview of the modifications mentioned in the review and changes in the levels of key regulators that mediate apoptosis. ↑ means upregulation, ↓ downregulation. *—modified miRNAs or its pri-miRNAs.

**Table 1 ijms-23-09272-t001:** Non-coding RNAs mentioned in the review.

ncRNA Name	Function in Glioblastoma	References
SOX2OT	Inhibition of apoptosis, increasing proliferation and TMZ resistance	[72,73,74]
FOXM1-AS	ALKBH5-mediated increasing of FOXM1 in GSCs	[31]
28S rRNA	Unmethylated C3872 decreased general protein level	[58]
JPX	Mediation of apoptosis by promoting FTO/PDK1 interaction	[70]
MALAT1	Activation of transcriptional factor NF-kB	[77]
AL080276.2	Unknown	[80]
AC092111.1	Unknown	[80]
SOX21-AS1	Promoting cell proliferation, apoptosis, migration and invasion	[79]
DNAJC9-AS1	Unknown	[80]
AC025171.1	Unknown	[80]
AL356019.2	Unknown	[80]
AC017104.1	Unknown	[80]
AC099850.3	Unknown	[80]
UNC5B-AS1	Unknown	[80]
AC006064.2	Unknown	[80]
AC010319.4	Unknown	[80]
AC016822.1	Unknown	[80]
SNORD44	Putative tumor suppressor	[88,90,91]
SNORD47	Inhibition of EMT by suppressing nuclear translocation and β-catenin activation, enhancing glioma temozolomide sensitivity	[82]
SNORD76	Regulation the levels of cyclin D1 and p21 protein. Retinoblastoma (Rb)-associated cell cycle arrest in S phase	[89]
GAS5	Putative tumor suppressor	[15,91]
lncGRS-1	Selectively inhibition of growth rate and increasing radiation sensitivity of tumor cells but not normal cells	[78]
miR-22 *	Inhibits the proliferation, motility, and invasion of glioblastoma cells	[100]
miR-376a-5p *	Inhibition of glioma proliferation and angiogenesis by regulating YAP1/VEGF signaling via targeting of SIRT1	[104]
miR-221/222 *	Promotes proliferation and migration of glioblastoma cells	[101]
miR-21 *	Promotes proliferation and migration of glioblastoma cells	[101]
miR-589-3p *	Promotes cell invasion	[101]
miR-125b *	In pediatric low grade glioma mature miR-125b inhibits growth and invasion, induces apoptosis	[105]
miR-1246 *	Reduction of PTEN or/and activating the PI3K/Akt/mTOR pathway	[71,101]
miR-25-3p *	Reduction of PTEN or/and activating the PI3K/Akt/mTOR pathway	[71,101]
miRNA-92 *	Reduction of PTEN or/and activating the PI3K/Akt/mTOR pathway	[71,101]
miR-126-5p *	Reduction of PTEN or/and activating the PI3K/Akt/mTOR pathway	[71,101]
miR-181a-5p *	Mature miR-181a-5p enhances apoptosis	[106]
miRNA-15b	Induction of apoptosis and inhibition stem characteristics	[110]
miRNA-29b	Induction of apoptosis and inhibition stem characteristics	[110]

*—modified miRNAs.

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
