# Peer review of "Post-Transcriptional Modifications of RNA as Regulators of Apoptosis in Glioblastoma"

_ijms, 2022, doi:10.3390/ijms23169272_

Round 1
Reviewer 1 Report
Manuscript is acceptable for publication in IJMS in the present form. Introduced supplementations, particularly Table 1. summarizing informations concerning all non-coding RNAs included in the review, as well as added definitions of majority used abreviations, makes manusript more easy to assimilate.
Author Response
Response to Reviewer 1 Comments
We thank reviewer for a thorough revision of our manuscript. Please find below our response to reviewer’s comments.
Point 1:
Manuscript is acceptable for publication in IJMS in the present form. Introduced supplementations, particularly Table 1. summarizing information concerning all non-coding RNAs included in the review, as well as added definitions of majority used abbreviations, makes manuscript more easy to assimilate.
Response 1:
We thank the reviewer for his thoughtful and thorough review and believe that his contribution was invaluable.
We highlighted (red highlighted text) all changes made when revising the manuscript to make it easier for the Editors to give a prompt decision on manuscript.
We thank the editors for considering our work for publication.
Yours faithfully,
Dymova Maya.
Reviewer 2 Report
The authors comprehensively overview RNA modifications related to apoptosis in high grade glioma. Some comments:
(.) Modomics is introduced, still other databses and resources could also be mentioned.
(.) The previously only figure presented in this review summarizes main apopototic signaling relevant for glioma cells, although the main focus is set on possible modifications or even hypothesized connections of RNA regulators to the depicted signaling excerpt, possibly as additional or refined subfigure. The now introduced second figure improves the manuscript!
(.) The structure of summarizing the various types of modifications could be refined, s.a. using numbering for paragraphs of the 3 main types of regulators of writers, erasers and readers of m6A. Since sofar the role of other RNA modifications is less known it makes sense to summarize some of them, still, a more consistent way of structuring and numbering as well as naming the several subsections would be more comprehensible, including the numbering of subsubsections.
(.) In the first round I would have suggested the conclusion to summarize the most promising targets for a directed therapy (partly mentioned before), or other hypothetised candidates if applicable. This has been somehow already refined!
Much success for ongoing and future work!
Author Response
Response to Reviewer 1 Comments
We thank reviewer for a thorough revision of our manuscript. Please find below our response to reviewer’s comments.
The authors comprehensively overview RNA modifications related to apoptosis in high grade glioma. Some comments:
Point 1.
(.) Modomics is introduced, still other databases and resources could also be mentioned.
Response 1
Corrected. We added more databases in addition to MODOMICS: RMBase v2.0 and SnOPY (A Small Nucleolar RNA Orthological Gene Database). Articles devoted to these databases are in the References under numbers 1-3.
Point 2.
(.) The previously only figure presented in this review summarizes main apoptotic signaling relevant for glioma cells, although the main focus is set on possible modifications or even hypothesized connections of RNA regulators to the depicted signaling excerpt, possibly as additional or refined subfigure. The now introduced second figure improves the manuscript!
Response 2
Thanks to your comments, the work really looks more complete and harmonious.
Point 3.
(.) The structure of summarizing the various types of modifications could be refined, s.a. using numbering for paragraphs of the 3 main types of regulators of writers, erasers and readers of m6A. Since so far the role of other RNA modifications is less known it makes sense to summarize some of them, still, a more consistent way of structuring and numbering as well as naming the several subsections would be more comprehensible, including the numbering of subsubsections.
Response 3
Corrected. We added numbering of paragraphs of m6A writers, erasers and readers in the subsections of m6A modifications. Moreover, we added numbering of the subsubsections in “Other modifications” subsection, a new subsubsection “m7G modification”, and a few words about uridine-to-pseudouridine modification in addition to snoRNA description. We believe that it is unnecessary to devote separate paragraphs to writers, erasers, and readers of modifications other than m6A, since there is much less information about each of these groups. It seems to us an optimal way of presenting information about these modifications in text without dividing into small subsections.
Point 4.
(.) In the first round I would have suggested the conclusion to summarize the most promising targets for a directed therapy (partly mentioned before), or other hypothetised candidates if applicable. This has been somehow already refined!
Much success for ongoing and future work!
Response 4
Thank you for your attention to our work, thanks to your comments, the work really looks more complete and harmonious. We hope that the work will be of interest to the readers of the journal and will be useful for understanding the complex processes of glioblastoma oncogenesis.
We highlighted (red highlighted text) all changes made when revising the manuscript to make it easier for the Editors to give a prompt decision on manuscript.
We thank the editors for considering our work for publication.
Yours faithfully,
Dymova Maya.
This manuscript is a resubmission of an earlier submission. The following is a list of the peer review reports and author responses from that submission.
Round 1
Reviewer 1 Report
Title
Post-transcriptional modifications of RNA as regulators of apoptosis in glioblastoma
Concise Summary and Comments
The authors review the epigenetic alterations of glioblastoma (GBM) in relation with apoptosis which are considered as promising markers for GBM. It is the importance of post-transcriptional control of gene expression in the development and progression of brain tumors. RNA modifications, especially N6-methyladenosine (m6A), and their RNA-modifying proteins such as methyltransferase like 3 (METTL3), α-ketoglutarate-dependent dioxygenase alkB homolog 5 (ALKBH5) and others, have also emerged as important epigenetic mechanisms for the aggressiveness and malignancy of GBM. The authors conclude that RNA modifications play increasingly important roles as regulators of tumor cells apoptosis.
The main problem with article is that there are recent quality reports about this issue (PMID: 32244981; PMID: 35625706; PMID: 32375856). This review does not give a more complete information in respect what has already been published.
A point of criticism is that the authors introduce new information and references in Conclusion (lines 444-57). However, it should not be accepted, because in this part of the article the authors should just give the final message of the manuscript.
Final decision
The article is generally well written and the considerations about the issue are consistent. However, this review does not provide quality information regarding what has been published.
Author Response
We thank reviewer for a thorough revision of our manuscript. Please find below our response to reviewer’s comments.
Point 1:
The main problem with article is that there are recent quality reports about this issue (PMID: 32244981; PMID: 35625706; PMID: 32375856). This review does not give a more complete information in respect what has already been published.
Response 1:
In our review, we tried to focus not on a comprehensive description of all possible RNA modifications, but on the subtle relationship between post-transcriptional bases modifications in RNA and the regulation of apoptosis in glioblastoma. Moreover, in our review we included the post transcriptional modifications of microRNAs, the subsection about SNORD RNAs and lncRNAs which are directly or indirectly involved in RNA modifications. We attempted to summarize all experimentally confirmed relationships between pathways and key regulators of apoptosis and post-transcriptional RNA modifications, as we believe that more intensive research is needed in this promising area.
Point 2:
A point of criticism is that the authors introduce new information and references in Conclusion (lines 444-57). However, it should not be accepted, because in this part of the article the authors should just give the final message of the manuscript.
Response 2:
This new information is intended to attract the attention of researchers. We have outlined the field for further research, indicating the most promising, in our opinion, ways of developing this direction. So the lines you've noticed are a kind of "future perspective". We see nothing against the requirements of the journal for authors to have this text in the final part of the article.
Corrected text is displayed in red color.
We highlighted (red highlighted text) all changes made when revising the manuscript to make it easier for the Editors to give a prompt decision on manuscript.
Yours faithfully,
Dymova Maya.
Reviewer 2 Report
The review is well-written and the authors provide a good summary of the advancements in our understanding of the post transcriptional modifications. However, the link of the PTM of RNAs to apoptosis in glioblastoma is still weak in terms of direct correlation by experimental evidence.
Some suggestions to improve the paper:
1) include a schematic diagram to explain the writers/erasers/readers RNA PTMs
2) A table listing the non-coding RNAs implicated in glioblastoma.
3) Recently lncGRS ( lncRNA) has been reported to have promising results in sensitizing glioblastoma cells towards radiotherapy " CRISPRi-based radiation modifier screen identifies long non-coding RNA therapeutic targets in glioma"Genome biology 21 (1), 1-18
Author Response
We thank reviewer for a thorough revision of our manuscript. Please find below our response to reviewer’s comments.
Point 1:
Include a schematic diagram to explain the writers/erasers/readers RNA PTMs
Response 1:
Corrected. We have added the figure 2 summarizing the relationship between RNA modifiers and apoptosis in glioblastoma.
Point 2:
A table listing the non-coding RNAs implicated in glioblastoma.
Response 2:
Corrected. We have added a table with the non-coding RNAs mentioned in the review.
Point 3:
Recently lncGRS (lncRNA) has been reported to have promising results in sensitizing glioblastoma cells towards radiotherapy " CRISPRi-based radiation modifier screen identifies long non-coding RNA therapeutic targets in glioma"Genome biology 21 (1), 1-18
Response 3:
Thanks for pointing this out. We have added information about lncGRS to the section on long non-coding RNAs.
Corrected text is displayed in red color.
We highlighted (red highlighted text) all changes made when revising the manuscript to make it easier for the Editors to give a prompt decision on manuscript.
Yours faithfully,
Dymova Maya.
Reviewer 3 Report
Authors undertaken the effort to summarize the latest information on post-transcriptional modifications of specific regulatory RNAs controlling process of apoptosis, paricularly in in human glioblastoma cells as potential target candidates for the de velopment of new therapies of that deadly cancer. Manuscript is very well written and include references concerning all known molecular mechanisms recognized as regulators of epitranscriptome modifications of individual target genes. Manuscript is acceptable for publication in IJMS with very minor comment that authors used many abbreviations without explanation, particularly in part 4. „The main regulators of apoptosis in glioblastoma cells”. Definition of that certain abbreviations will make the text easier for readers to acquire.
Author Response
We thank reviewer for a thorough revision of our manuscript. Please find below our response to reviewer’s comments.
Point 1:
Manuscript is acceptable for publication in IJMS with very minor comment that authors used many abbreviations without explanation, particularly in part 4. „The main regulators of apoptosis in glioblastoma cells”. Definition of that certain abbreviations will make the text easier for readers to acquire.
Response 1:
We have added definitions of major regulators mentioned in our review in part 4.
Corrected text is displayed in red color.
We highlighted (red highlighted text) all changes made when revising the manuscript to make it easier for the Editors to give a prompt decision on manuscript.
Yours faithfully,
Dymova Maya.
Reviewer 4 Report
In this review the effect of RNAs and their post-transcriptional modifications are summarized in the focus of glioblastoma. The review is well written, however it has some limitations.
Little information is presented about microRNAs. Due to the fact that these are the most known non-coding RNA molecules it would be important to present more details about their role in the development of glioblastoma.
The manuscript contains only 1 Figure. 1 other Figure that summarizes the RNA modification processes would be helpful.
A table is also necessary about the identified RNA molecules involved in glioblastoma.
In the conclusion section some clinical applications (e.g. RNAs in therapy) should be mentioned. It would be also interesting to see how modifying RNA molecules can support their therapeutic application.
Author Response
We thank reviewer for a thorough revision of our manuscript. Please find below our response to reviewer’s comments.
Point 1:
Little information is presented about microRNAs. Due to the fact that these are the most known non-coding RNA molecules it would be important to present more details about their role in the development of glioblastoma.
Response 1:
Corrected. We have added a subsection about role of microRNAs and their modifications in glioblastoma.
Point 2:
The manuscript contains only 1 Figure. 1 other Figure that summarizes the RNA modification processes would be helpful.
Response 2:
Corrected. We have added figure to summarize RNA modifications, related molecules and apoptosis.
Point 3:
A table is also necessary about the identified RNA molecules involved in glioblastoma.
Response 3:
Corrected. We have added a table with ncRNA involving in RNA modification-dependent regulation of apoptosis.
Point 4:
In the conclusion section some clinical applications (e.g. RNAs in therapy) should be mentioned. It would be also interesting to see how modifying RNA molecules can support their therapeutic application.
Response 4:
Corrected. We added the sentences in the conclusion section.
Corrected text is displayed in red color.
We highlighted (red highlighted text) all changes made when revising the manuscript to make it easier for the Editors to give a prompt decision on manuscript.
Yours faithfully,
Dymova Maya.
Reviewer 5 Report
The authors overview RNA modifications related to apoptosis in high grade glioma. Some remarks:
(.) Modomics is introduced but other databses and resources could also be mentioned for the sake of comprehensiveness.
(.) The only figure presented in this review summarizes main apopototic signaling relevant for glioma cells, although the main focus should be set on possible modifications or even hypothesized connections of RNA regulators to the depicted signaling excerpt, possibly as additional or refined subfigure.
(.) The structure of summarizing the various types of modifications could be refined, s.a. using numbering for paragraphs of the 3 main types of regulators of writers, erasers and readers of m6A.
Since sofar the role of other RNA modifications is less known it makes sense to summarize some of them, still, a more consistent way of structuring and numbering as well as naming the several subsections would be more comprehensible, including the numbering of subsubsections.
(.) The conclusion could summarize the most promising targets for a directed therapy (partly mentioned before), or other hypothetised candidates if applicable.
Author Response
We thank reviewer for a thorough revision of our manuscript. Please find below our response to reviewer’s comments.
Point 1:
Modomics is introduced but other databses and resources could also be mentioned for the sake of comprehensiveness.
Response 1:
Corrected. We have mentioned RNAMDB and RMBase v2.0 in addition to MODOMICS.
Point 2:
The only figure presented in this review summarizes main apopototic signaling relevant for glioma cells, although the main focus should be set on possible modifications or even hypothesized connections of RNA regulators to the depicted signaling excerpt, possibly as additional or refined subfigure.
Response 2:
Corrected. We have added Figure 2. summarizing modifications mentioned in our review and key regulators of these modifications.
Point 3:
The structure of summarizing the various types of modifications could be refined, s.a. using numbering for paragraphs of the 3 main types of regulators of writers, erasers and readers of m6A.
Since sofar the role of other RNA modifications is less known it makes sense to summarize some of them, still, a more consistent way of structuring and numbering as well as naming the several subsections would be more comprehensible, including the numbering of subsubsections.
Response 3:
Corrected. We have added numbering of the subsubsections in “Other modifications” subsection, a new subsubsection “m7G modification”, and a few words about uridine-to-pseudouridine modification in addition to snoRNA description.
Point 4:
The conclusion could summarize the most promising targets for a directed therapy (partly mentioned before), or other hypothetised candidates if applicable.
Response 4:
Corrected. We have added in conclusion promising targets mentioned before and tools for further research.
Corrected text is displayed in red color.
We highlighted (red highlighted text) all changes made when revising the manuscript to make it easier for the Editors to give a prompt decision on manuscript.
Yours faithfully,
Dymova Maya.
Round 2
Reviewer 1 Report
Final decision
In conclusion, I appreciate the modifications that the authors brought to their manuscript according to the reviewer comments. The authors have made several changes in the manuscript, which has resulted in an improved new version of the article. As it was told, this article does not go give relevant information on previous retrospective published surveys on this issue. Regrettably, the reviewer considerations have been partially solved by the authors.